# Screening of a *Saccharomyces cerevisiae* Strain with High 3-Methylthio-1-Propanol Yield and Optimization of Its Fermentation Conditions

**DOI:** 10.3390/foods13091296

**Published:** 2024-04-23

**Authors:** Qi Sun, Jinghao Ma, Rana Abdul Basit, Zhilei Fu, Xiaoyan Liu, Guangsen Fan

**Affiliations:** 1Key Laboratory of Geriatric Nutrition and Health, Beijing Technology and Business University, Ministry of Education, Beijing 100048, China; s17321914603l@163.com (Q.S.); 15032049018@163.com (J.M.); basit98ft@gmail.com (R.A.B.); liuxiaoyan@btbu.edu.cn (X.L.); 2School of Biology and Food Science, Hebei Normal University for Nationalities, Chengde 067000, China; 13833415159@163.com; 3China Food Flavor and Nutrition Health Innovation Center, School of Food and Health, Beijing Technology and Business University, Beijing 100048, China; 4Sweet Code Nutrition & Health Institute, Zibo 256306, China

**Keywords:** 3-methylthio-1-propanol, *Saccharomyces cerevisiae*, strain identification, fermentation condition optimization, response surface methodology

## Abstract

3-Methylthio-1-propanol (3-Met) is an important flavor compound in various alcoholic beverages such as *Baijiu* and *Huangjiu*. To maintain the content of 3-Met in these alcoholic beverages, it is necessary to screen a micro-organism with high yield of 3-Met from the brewing environment. In this study, the ability of yeast strains from the *Baijiu* brewing to produce 3-Met was analyzed, aiming to obtain yeast with high-yield 3-Met, and its fermentation conditions were optimized. Firstly, 39 yeast strains were screened using 3-Met conversion medium. The results showed that the majority of the strains from *Baijiu* brewing sources could produce 3-Met, and nearly half of the strains produced more than 0.5 g/L of 3-Met. Among these, yeast F10404, Y03401, and Y8#01, produced more than 1.0 g/L of 3-Met, with yeast Y03401 producing the highest amount at 1.30 g/L. Through morphological observation, physiological and biochemical analysis, and molecular biological identification, it was confirmed that yeast Y03401 was a *Saccharomyces cerevisiae*. Subsequently, the optimal fermentation conditions for 3-Met production by this yeast were obtained through single-factor designs, Plackett–Burman test, steepest ascent path design and response surface methodology. When the glucose concentration was 60 g/L, yeast extract concentration was 0.8 g/L, L-methionine concentration was 3.8 g/L, initial pH was 4, incubation time was 63 h, inoculum size was 1.6%, shaking speed was 150 rpm, loading volume was 50 mL/250 mL, and temperature was 26 °C, the content of 3-Met produced by *S. cerevisiae* Y03401 reached a high level of 3.66 g/L. It was also noteworthy that, in contrast to other study findings, this yeast was able to create substantial amounts of 3-Met even in the absence of L-methionine precursor. Based on the clear genome of *S. cerevisiae* and its characteristics in 3-Met production, *S. cerevisiae* Y03401 had broad prospects for application in alcoholic beverages such as *Baijiu*.

## 1. Introduction

Pineapple alcohol, or 3-Methylthio-1-propanol (3-Met), is a sulfur-containing compound that imparts pleasant meaty, grilled cheese, sweet, onion, or vegetable flavors to alcoholic beverages such as *Baijiu*, *Huangjiu*, wine, rice wine, beer, brandy, and whiskey [1,2,3,4,5,6,7]. It is an important flavor compound in these alcoholic beverages. For example, in soy-flavored *Baijiu*, 3-Met imparts a ripe potato aroma [3]; in *Magli* wine, it imparts a unique vegetable aroma [4]; in beer and wine, it contributes a unique cooked potato or vegetable flavor [8,9]; in fruit wines such as cider, it is a major flavor compound that has a significant impact on product quality [6,10]. Furthermore, *Huangjiu* has a strong 3-Met fragrance, particularly in the refreshing variety, which gives *Huangjiu* a cooked vegetable odor [11]. In conclusion, 3-Met has a significant impact on the quality of alcoholic beverages due to its concentration.

Studies have shown that the major source of 3-Met in alcoholic beverages is the metabolic activity of relevant micro-organisms during the brewing process. The principal source was yeast, which produced 3-Met by metabolizing basic materials via the Ehrlich route or precursor chemicals from other microbial metabolisms [12]. Unfortunately, at present, these alcohol products generally have a low content of 3-Met during the brewing process, which affects the quality of the related products to a certain extent [12]. The lack or almost complete absence of critical strains that generated significant amounts of 3-Met during the brewing process was one of the main causes of the aforementioned process. In reported studies, except for *Kluyveromyces lactis* KL71, *Hyphopichia burtonii* YHM-G, *S. cerevisiae* SC408, *S. cerevisiae* SC57, *S. cerevisiae* S288C and *Saccharomycopsis fibuligera* Y1402, other natural strains produced less than 0.5 g/L of 3-Met under optimal fermentation conditions [13,14,15,16,17,18,19,20,21]. Modern genetic technology and genetic modification have made it entirely possible to increase natural strains’ capacity to produce high levels of 3-Met; however, Chinese laws and regulations currently prohibit the use of genetically modified strains in the production of traditional brewing food, which raises concerns among consumers regarding the safety of brewing food [22,23,24]. Therefore, screening high-yield 3-Met strains from the production environment of brewing food and applying them appropriately to the brewing of alcoholic beverages was an effective way to address the above issues.

Currently, multiple yeast strains that produced 3-Met had been screened from the brewing environment of traditional brewing foods, such as *S. cerevisiae* [25], *Zygosaccharomyces rouxii* [1], *K. lactis* [16], *Geotrichum candidum* [26], *H. burtonii* [21] and *S. fibuligera* [20]. However, except for *H. burtonii* and *S. fibuligera*, which had a higher ability to produce 3-Met, the levels of 3-Met produced by other strains were generally low, which cannot meet the needs of products for 3-Met content. Therefore, it was necessary to further screen relevant functional micro-organisms from brewing foods, especially the production environment of alcoholic beverage products, in order to use these micro-organisms to increase the content of 3-Met in alcoholic beverages and improve their quality.

## 2. Materials and Methods

### 2.1. Materials

The experiment’s yeast strains were selected from several aroma-type *Baijiu* brewing environments and then preserved by our research team. L-methionine (L-Met), 3-Met standard, and chromatographic grade methanol were purchased from Sigma (St. Louis, MO, USA). Unless otherwise stated, all other chemicals were analytical grade and commercially available.

### 2.2. Medium

Yeast peptone dextrose (YPD) medium: yeast extract 10 g/L, peptone 20 g/L, glucose 20 g/L, sterilized at 121 °C for 20 min.

3-Met conversion medium: glucose 30 g/L, yeast extract 0.8 g/L, L-Met 4 g/L, KH_2_PO_4_ 8 g/L, K_2_HPO_4_ 6 g/L, NaCl 2 g/L, ZnSO_4_ 0.03 g/L, MgCl 0.01 g/L, FeCl_2_ 0.02 g/L (through a 0.22 μm membrane), pH 5.0, sterilized at 121 °C for 20 min.

Wallerstein laboratory nutrient (WL) medium: yeast extract 5 g/L, tryptone 5 g/L, glucose 50 g/L, agar 20 g/L, KH_2_PO_4_ 0.55 g/L, KCl 0.425 g/L, CaCl_2_ 0.125 g/L, FeCl_3_ 0.0025 g/L, MgSO_4_ 0.125 g/L, MnSO_4_ 0.0025 g/L, bromocresol green 0.022 g/L, pH 6.5, sterilized at 121 °C for 20 min.

Hydrogen sulfide test medium: NaCl 5 g/L, tryptone 25 g/L, gelatin 120 g/L, beef extract 7.5 g/L, 10% FeSO_4_·7H_2_O (sterilized through a 0.22 μm microporous membrane) 5 mL, pH 7.0, sterilized at 115 °C for 20 min.

Indole test medium: NaCl 5 g/L, tryptone 10 g/L, pH 7.8, sterilized at 121 °C for 20 min.

Methyl red test medium: K_2_HPO_4_ 2 g/L, glucose 5 g/L, tryptone 5 g/L, pH 7.0–7.2, sterilized at 112 °C for 30 min.

Voges–Proskauer test medium: NaCl 5 g/L, tryptone 10 g/L, pH 7.8, sterilized at 121 °C for 20 min. 

Citrate test medium: NH_4_H_2_PO_4_ 1 g/L, NaCl 5 g/L, K_2_HPO_4_ 1 g/L, sodium citrate 2 g/L, agar 20 g/L, MgSO_4_ 0.2 g/L, pH 6.8, sterilized at 121 °C for 20 min. 

Starch hydrolysis test medium: NaCl 5 g/L, beef extract 5 g/L, agar 20 g/L, tryptone 10 g/L, soluble starch 2 g/L, sterilized at 121 °C for 20 min. 

Urea medium: tryptone 1 g/L, agar 15 g/L, NaCl 5 g/L, KH_2_PO_4_ 2 g/L, sterilized at 121 °C for 20 min. 20 g/L urea (sterilized through a 0.22 μm microporous membrane). 

Gelatin hydrolysis experimental medium: NaCl 5 g/L, tryptone 10 g/L, gelatin 120 g/L, beef extract 3 g/L, pH 7.2–7.4, sterilized at 121 °C for 20 min. 

Skim milk test medium: Dissolve 10.167 g of commercial skim milk medium in 100 mL of water, sterilize at 115 °C for 15 min. 

Sugar fermentation medium: NaCl 5 g/L, tryptone 10 g/L, 0.2% bromocresol purple, pH 7.8, sterilized at 115 °C for 20 min. 

Carbon assimilation medium: beef extract 5 g/L, tryptone 10 g/L, NaCl 3 g/L, Na_2_HPO_4_ 2 g/L, sugar or alcohol 10 g/L, bromothymol blue 0.03 g/L, sterilized at 115 °C for 20 min. 

Nitrogen assimilation medium: MgSO_4_·7H_2_O 0.05 g/L, glucose 2 g/L, KH_2_PO_4_ 0.1 g/L, pH 5–6, nitrogen source 1 g/L, sterilized at 115 °C for 15 min.

### 2.3. Screening of High-Yielding 3-Met Yeast Strains

The yeast strains preserved in glycerin tube at −20 °C were inoculated into YPD medium and activated for 48 h at 28 °C and 180 rpm. The activated yeast was then inoculated into 3-Met conversion medium with a 1% inoculum size and incubation for 48 h at 28 °C and 180 rpm. After centrifugation at 4 °C and 13,751× *g*, the supernatant was collected, and the 3-Met content was determined using high-performance liquid chromatography (HPLC). 

### 2.4. Identification and Performance Testing of Yeast Strains

The high-yielding 3-Met yeast strains were identified according to Ma et al., Buchana et al. and Dong et al. [21,27,28], and their growth temperature range (20, 25, 30, 35, 40, 45, and 50 °C), pH range (1–14), glucose tolerance (4.8, 9.1, 13.0, 16.7, 20, 23.1, 25.9, 28.6, 31.0, 33.3, 35.5, 37.5, 39.4, 41.2, 42.9, 44.4, 45.9, 47.4, 48.7, and 50.0%, *w*/*w*), NaCl tolerance (0, 3, 6, 8, 11, 13, 15, 17, 19, and 21%, *w*/*v*), ethanol tolerance (0, 3, 6, 9, 12, 15, 18, and 21%, *v*/*v*), and 3-Met tolerance (1, 2, 3, 4, 5, 6, 7, 8, 9, 10, 11, 12, 13, 14, and 15 g/L) were determined using liquid culture methods. The cell density after 36 h of culture was measured using turbidity, and the absorbance values at 560 nm of different samples were determined with the corresponding blank control of the culture medium without inoculation. 

### 2.5. Optimization of 3-Met Production by Single-Factor Design (SFD)

SFD was conducted to explore the effects of medium components (glucose concentration, yeast extract concentration, L-Met concentration, surfactant type and concentration) and fermentation conditions (initial pH, temperature, incubation time, inoculum size, shaking speed, and loading volume) on 3-Met production by *S. cerevisiae* Y03401. The important factors affecting 3-Met yield and their corresponding optimal levels were determined (Table 1).

### 2.6. Optimization of 3-Met Production by Plackett–Burman (PB) Test

Based on the results of SFD, nine factors including glucose concentration (X_1_), yeast extract concentration (X_2_), L-Met concentration (X_3_), temperature (X_4_), initial pH (X_5_), incubation time (X_6_), shaking speed (X_7_), loading volume (X_8_), and inoculum size (X_9_) were selected for the PB test to further determine the main factors influencing 3-Met yield. This was performed in order to subsequently conduct response surface methodology (RSMD) to obtain the optimal fermentation conditions (Table 2).

### 2.7. Optimization of 3-Met Production by Steepest Ascent Path Design (SAPD)

Based on the results of the PB test, factors significantly affecting 3-Met production were selected for the SAPD to obtain a reasonable range for the design of RSMD. The direction of each variable in the experiment was determined based on the regression results of the PB test, and the step size for each variable was determined based on the regression model of the PB test and actual experience. The specific design was shown in Table 3.

### 2.8. Optimization of 3-Met Production by Response Surface Methodology (RSMD)

Based on the determination of the center point from the SAPD, an RSMD was conducted to optimize three factors: incubation time (A), L-Met concentration (B), and yeast extract concentration (C). According to the Box–Behnken experimental design (BBD, Design-Expert Software 11.0, StatEase Inc., Minneapolis, MN, USA), a total of 17 experiments were designed (including five repetitions of the center point) (Table 4).

### 2.9. Determination of 3-Met Content

After centrifugation at 13,751× *g* for 10 min, the supernatant of the fermentation broth was filtered through a 0.22 μm membrane, and then appropriately diluted with pure water for HPLC detection. The HPLC method followed the parameters described by Ma et al. [21], as follows: C-18 reverse-phase column (ZORBAX Eclipse Plus C-18, 4.6 × 250 mm, 5 μm); mobile phase of methanol:water = 10:90, with pure phosphoric acid to adjust the pH to 3; flow rate: 0.7 mL/min; detection wavelength 215 nm; column temperature 30 °C; injection volume 10 μL.

### 2.10. Data Analysis

A one-way ANOVA (*p* < 0.05) with Tukey’s test was used to analyze statistical differences in the assessed methodologies. Each experiment was conducted in triplicate, and the experimental data were processed and plotted using Excel 2019, SPSS 24.0, and Design-Expert 11.

## 3. Results and Discussion

### 3.1. Screening of High-Yielding 3-Met Strain

A total of 39 strains that were kept in our lab were separately activated and inoculated into a 3-Met conversion medium. Their 3-Met production was assessed during a 48 h fermentation period. The findings are shown in Table 5, which demonstrates that most of the strains were capable of producing 3-Met. Only eight of the examined strains failed to produce any discernible 3-Met. Moreover, several strains of the yeasts that produce 3-Met showed differing capacities for the manufacture of 3-Met. Nearly half of the strains produced 3-Met at concentrations exceeding 0.5 g/L, which was unexpectedly high compared to previous research reports. This might be related to the fact that most of the selected strains originated from brewing environments of sesame-flavored *Baijiu*. Notably, among the tested strains, three strains produced significantly higher levels of 3-Met, exceeding 1.0 g/L, namely the strains F10404, Y03401, and Y8#01, with yield of 1.21 g/L, 1.30 g/L, and 1.10 g/L, respectively. Remarkably, strain Y03401 was chosen for further investigation because it showed the highest 3-Met output among all the strains examined.

### 3.2. Identification and Performance Testing of Strain Y03401

Strain Y03401 produced blue–green colonies that were elevated and had smooth edges, but the colonies’ centers remained blue–green after growing on WL medium (Figure 1a). The surface of the colony was smooth, sticky, wet, and pickable. Under a microscope, the cells showed conventional yeast features: they were round or elliptical, budding at one end, and devoid of mycelium (Figure 1b).

Analysis of its physiological and biochemical characteristics (Table 6) revealed that yeast Y03401 could grow when D-xylose, fructose, glucose, maltose, lactose, and sucrose were used as the sole carbon sources. It produced acid and gas when fructose, glucose, maltose, and sucrose were the sole carbon sources, while only producing gas, not acid, when D-xylose and lactose were the sole carbon sources. In carbon source assimilation tests, yeast Y03401 could grow when ethanol, glycerol, and inulin were used as the sole carbon sources but not when D-ribose was used. In nitrogen assimilation tests, urea, potassium nitrate, ammonium sulfate, L-Met, and L-tyrosine all promoted the growth of yeast Y03401 as sole nitrogen sources. Additionally, yeast Y03401 showed poor results in the tests for hydrogen sulfide, indole, Voges–Proskauer, urea hydrolysis, and gelatin liquefaction, but it showed positive results in the tests for methyl red, citrate utilization, and starch hydrolysis. This showed that yeast Y03401 could thrive when sodium citrate was the only carbon source and created amylase to break down starch. Y03401 may ferment glucose and produce acidic chemicals such as pyruvic acid, lactic acid, succinic acid, acetic acid, or formic acid. After extracting the whole DNA from yeast Y03401, the 26s rDNA gene fragment was successfully amplified using specific primers, NL1 (5′-GCATATCAATAAGCGGAGGAAAAG-3′) and NL4 (5′-GGTCCGTGTTTCAAGACGG-3′). This yeast’s identity was verified by comparing and analyzing the gene sequence in the NCBI database, which revealed a 100% similarity to the *S. cerevisiae* source. It was noteworthy, nevertheless, that this particular yeast’s physiological and biochemical traits did not match those of *S. cerevisiae* Y1914 and Y3401, which could be because of variations in their living conditions leading to partial variations in their physiological and biochemical traits [29,30]. 

Experimental results indicated that the highest growth temperature for *S. cerevisiae* Y03401 was 40 °C, and its optimal growth temperature was 25 °C, suitable for the brewing of *Baijiu*, *Huangjiu*, and other wine products [31]. *S. cerevisiae* Y03401 had low tolerance to NaCl, only 6%, although it cannot be used in some high-salt-concentration foods, it adapted well to the production of alcohol-containing products. *S. cerevisiae* Y03401 had moderate ethanol tolerance and grew well when the ethanol volume fraction is 6% or lower, which made it possible to adjust to the 6%-or-less ethanol content that is typically produced in the brewing of *Baijiu* and some other alcoholic beverages [32,33]. *S. cerevisiae* Y03401 tolerated a maximum glucose concentration of 45.9%, slightly higher than *H. burtonii* YHM-G and *Clavispora lusitaniae* YX3307, showing relatively high sugar tolerance [21,34]. It had broad pH tolerance and grew at pH 2–10, making it an excellent strain that can adapt to different environmental pH levels, with its optimal growth pH being 4.0, which might be related to its origin in the brewing of *Baijiu* [31]. In addition, *S. cerevisiae* Y03401 had a tolerance of 12 g/L to 3-Met, lower than *H. burtonii* YHM-G and *S. cerevisiae* CEN.PK113–7D [21,25]. In conclusion, *S. cerevisiae* Y03401 was quite suitable for use in *Baijiu*, *Huangjiu*, and other wine products.

### 3.3. Optimization of 3-Met Production by SFD

#### 3.3.1. Effect of Medium Components on 3-Met Production by *S. cerevisiae* Y03401

Glucose is a vital component of microbial growth, metabolism, and reproduction. It serves as both the energy supply and the carbon skeleton for microbial cell creation. However, since various microbes have varying glucose tolerances, higher glucose concentration was not always preferable for the development and metabolism of microbes; rather, it was helpful in specific circumstances [35]. The influence of glucose concentration on the production of 3-Met by *S. cerevisiae* Y03401 is clearly depicted in Figure 2a. The figure showed that when glucose concentration increased, the demand for energy and carbon sources necessary for the growth, metabolism, and reproduction of *S. cerevisiae* Y03401 escalated progressively. This increased the concentration of 3-Met that was produced. The concentration of 3-Met peaked between 60 and 90 g/L as the glucose content rose. However, further increasing the glucose concentration led to a decrease in 3-Met yield, as the high osmotic pressure caused by high glucose concentration affected the growth and metabolism of *S. cerevisiae* Y03401. Considering the yield and cost, the glucose concentration was chosen as 60 g/L, higher than the optimal glucose concentration reported in previous studies, which might be related to the higher glucose tolerance of this strain [17,21].

Previous studies have shown that the Ehrlich path was used by yeast to generate 3-Met. The presence of additional nitrogen sources in the medium would affect the yeast’s ability to convert the substrate L-Met into 3-Met. However, earlier studies showed that yeast could make 3-Met more effectively when it was exposed to an appropriate concentration of yeast extract. This might be due to the fact that yeast extract, in addition to serving as a nitrogen source, also contained numerous other nutritional components such as various vitamins, amino acids, and biotin, which were important growth factors for yeast’s growth and reproduction, thereby promoting its high yield of 3-Met [14,21]. As Figure 2b illustrates, the experimental groups with varying yeast extract quantities produced greater amounts of 3-Met in comparison to the control group that did not have any yeast extract added. *S. cerevisiae* Y03401 generated the highest concentration of 3-Met, particularly at a yeast extract concentration of 0.4 g/L. However, excessively high yeast extract concentrations would inhibit the production of 3-Met, which fully confirmed the conclusions reported in previous studies. Lower concentrations of yeast extract provided the necessary growth factors for yeast, while higher concentrations of yeast extract would exert a strong inhibitory effect on the Ehrlich pathway of yeast [21]. In contrast to other studies, *S. cerevisiae* Y03401 needed a considerably low optimum concentration of yeast extract in order to produce 3-Met. This might be because various strains of the yeast have somewhat different associated metabolic pathways [17,21].

L-Met is an essential precursor for yeast to produce 3-Met through the Ehrlich pathway, and its concentration is crucial for the synthesis of 3-Met by yeast [36]. The concentration of 3-Met produced by *S. cerevisiae* Y03401 initially increased and subsequently decreased in response to an increase in L-Met concentration, as shown in Figure 2c. The highest concentration of 3-Met was achieved at L-Met concentrations of 2–4 g/L. On the other hand, 3-methylthio-1-propanoic acid production might be excessive in yeast due to too-high L-Met concentrations, which would prevent yeast development and metabolism. The yeast would desulfurate L-Met to lower its effective concentration to counteract this inhibition, which would lower the amount of 3-Met it produced [37,38]. Contrary to earlier study findings, *S. cerevisiae* Y03401 was able to synthesize significant amounts of 3-Met with a yield more than 0.5 g/L even in the absence of L-Met. The reason for this might be because the strain can either manufacture 3-Met from glucose de novo or it can synthesize L-Met from glucose and utilize the Ehrlich route to make 3-Met. Additionally, for this reason, this strain’s optimal L-Met concentrations for generating 3-Met were somewhat lower than those of other strains [16,21].

According to earlier research, adding appropriate surfactants to the medium, including Tween and Triton, may aid in the production of 3-Met by the yeast. These surfactants may have enhanced the permeability of the cell membrane, the dissolved oxygen concentration, and the expression of enzymes involved in metabolism while the cells were being incubated [21,39]. However, in this study, the expected effect was not achieved (Figure 2d). Among the tested surfactants, none of them could promote the production of 3-Met by *S. cerevisiae* Y03401. On the contrary, they all had a certain inhibitory effect, especially glycerol and this might be due to differences in the composition of the cell membrane of yeast strains, leading to different effects of surfactants on their growth and metabolism.

#### 3.3.2. Effect of Fermentation Conditions on 3-Met Production by *S. cerevisiae* Y03401

The growth, reproduction, and metabolic pathways of micro-organisms are influenced by pH, which also plays a crucial regulatory role in the accumulation of essential metabolic products. pH influences the presence of nutrients in the microbial growth environment, the membrane potential and permeability of the microbial cells, and membrane protein function [40]. The results of the initial pH’s effect on the yield of 3-Met by *S. cerevisiae* Y03401 are shown in Figure 3a. It was observed that at an initial pH of 4.0, *S. cerevisiae* Y03401 produced the highest level of 3-Met, while lower or higher initial pH levels would affect its yield. This might be attributed to the fact that, under these initial pH conditions, the capacity of *S. cerevisiae* Y03401 to absorb nutrients or the functionality of enzymes and proteins involved in the Ehrlich pathway was relatively impaired, as evidenced by its optimal growth pH of 4.0. Additionally, this was similar to the optimal pH for 3-Met production by *H. burtonii* YHM-G, which also originated from *Baijiu*, and this might be related to the slightly acidic environment in which both were used for *Baijiu* production [21,29].

Temperature is an important parameter that affects the growth, reproduction, and metabolic pathways of micro-organisms. Suitable temperatures are conducive to the growth, reproduction, and accumulation of metabolic products in micro-organisms, while lower temperatures will affect the diffusion of nutrients and the fluidity of microbial cell membranes, leading to a slower rate of nutrient intake and lower enzymatic activity within the micro-organism, which is not conducive to its growth and reproduction. On the other hand, higher temperatures cause damage to the microbial cell membrane and loss of enzymatic activity within the organism, which ultimately hinders its growth and reproduction [41]. Based on the growth temperature range of *S. cerevisiae* Y03401, the production of 3-Met was investigated within the range of 22–38 °C. The results (Figure 3b) showed that with an increase in the temperature, the amount of 3-Met produced initially increased and then decreased. When the temperature was 26 °C, the yield reached its maximum, which was consistent with its optimal growth temperature and slightly lower than the reported optimal temperature for other yeast strains producing 3-Met [14,21].

The large accumulation of microbial metabolic products generally begins during the logarithmic growth phase, while the peak production mainly occurs during the stable growth phase and the pre-death phase [21]. Different micro-organisms often have different optimal fermentation periods for accumulating metabolic products. Therefore, in order to accumulate more 3-Met, it was necessary to study the fermentation process of *S. cerevisiae* Y03401 for producing 3-Met. The results (Figure 3c) showed that with the prolongation of incubation time, the content of 3-Met produced by *S. cerevisiae* Y03401 gradually increased. The yield reached its highest point at 48 h of fermentation, and then decreased slightly. This might be because the high concentration of 3-Met can have a toxic effect on yeast cells, and the yeast converted it into other metabolic products with weaker toxic effects for self-protection. Alternatively, it might be due to the lack of nutrients in the late stage of fermentation, and the yeast used 3-Met as a nutrient and metabolized it into other substances. It was worth noting that compared with *H. burtonii* YHM-G, this yeast accumulated a large amount of 3-Met in a shorter time (within 24 h), which might be related to the higher enzymatic activity of the Ehrlich pathway, the higher incubation time, or the ability of de novo synthesis [21].

In the fermentation process, the growth rate of the strain and the length of the fermentation period are closely related to the inoculum size. Generally, a properly larger inoculum size can improve the growth rate, shorten the fermentation period, and reduce production costs [21]. The quantity of 3-Met generated by *S. cerevisiae* Y03401 somewhat increased as the inoculum size increased, as Figure 3d illustrates. The highest yield occurred at 1.6% inoculum size, and when inoculum size increased, the output declined marginally. This may be the result of a larger inoculum size causing the fermentation system’s oxygen concentration to be inadequate and nutrients to be used more quickly, which hindered the yeast’s ability to produce the necessary metabolic products and undergo necessary transformations. However, overall, *S. cerevisiae* Y03401 produced a high yield of 3-Met under different inoculum sizes, which might be related to its good environmental adaptability and its ability to rapidly convert L-Met into 3-Met.

It has been shown in earlier research that substantial yields of 3-Met by yeast were possible in an oxygen-rich environment [16]. The two most direct elements influencing the oxygen concentration in a shaken flask fermentation system were the loading volume and the shaking speed. Therefore, it was crucial to investigate how these two variables influenced the capability of *S. cerevisiae* Y03401 to produce 3-Met. At lower shaking speeds (50 rpm), the flask’s oxygen level was insufficient, adversely affecting the growth and fermentation of *S. cerevisiae* Y03401, and subsequently reducing the production of 3-Met, as depicted in Figure 3e. The yield of 3-Met increased as the shaking speed increased because the oxygen concentration grew steadily. The maximum yield was obtained at 150–200 rpm shaking rates but when the shaking speed increased, the tremendous shear force brought on by these high speeds damaged the yeast cells to some degree, reducing the yeast’s capacity to ferment even while the oxygen concentration was rising. These results were consistent with the SFD optimization of shaking speed for *H. burtonii* YHM-G [21]. Furthermore, the yield of 3-Met produced by yeast also rose initially, then decreased, as loading volume increased (Figure 3f). It is probable that the combined effects of oxygen concentration and physical shear force led to the greatest production of 3-Met, which was seen at a loading volume of 50 mL/250 mL.

### 3.4. Optimization of 3-Met Production by PB Test

The concentration of glucose, yeast extract, L-Met, temperature, initial pH, incubation time, shaking speed, loading volume, and inoculum size were among the nine factors that were chosen for the PB test based on the SFD results. These factors were intended to shed additional light on the significant factors influencing the yield of 3-Met by *S. cerevisiae* Y03401. The fitting regression equation linking 3-Met yield to each element was derived by statistical analysis of the experimental data, and it looks like this:Y = 0.09 − 0.00491 × X_1_ + 1.620 × X_2_ + 0.1540 × X_3_ + 0.0079 × X_4_ − 0.0146 × X_5_ + 0.01602 × X_6_ + 0.00041 × X_7_ − 0.00503 × X_8_ + 0.1795 × X_9_

Through variance analysis and significance analysis (Table 7), it was found that the *F*-value of the fitted regression model was 8.46, with a corresponding *p*-value of 0.015 (*p* < 0.05), indicating the significance of the fitted model. The coefficient of determination *R*^2^ for the regression equation was 0.9384, indicating that the model had high accuracy and was used to predict the yield of 3-Met by *S. cerevisiae* Y03401. Additionally, the regression equation showed that loading volume, initial pH, and glucose concentration had a negative effect on the yield of 3-Met by *S. cerevisiae* Y03401, while the other six factors had a positive effect. The significance analysis indicated that yeast extract concentration, L-Met concentration, and incubation time had a significant impact on the yield of 3-Met, while the other factors had an insignificant impact. Therefore, further optimization was carried out for these three factors, while the levels of the other factors were set to the optimal levels determined by SFD designs, glucose concentration of 60 g/L, initial pH of 4.0, temperature of 26 °C, inoculum size of 1.6%, shaking speed of 150 rpm, and loading volume of 50 mL/250 mL.

### 3.5. Optimization of 3-Met Production by SAPD

The results of the SAPD showed that the yield of 3-Met produced by *S. cerevisiae* Y03401 matched the expected values. As the three significant factors progressed in the direction of increasing their influence on the response value, the yield of 3-Met gradually reached its peak and then showed a decreasing trend. This led to the determination of the central values for the optimal optimization range for subsequent RSMD: yeast extract concentration of 0.8 g/L, incubation time of 60 h, and L-Met concentration of 3 g/L. Under the above fermentation conditions, the yield of 3-Met by *S. cerevisiae* Y03401 could reach 3.29 g/L.

### 3.6. Optimization of 3-Met Production by RSMD

In a multi-component environment, particularly when there are interactions between the variables, the RSMD can efficiently ascertain the ideal degree of impact of each element and the interactions between the factors, swiftly establishing the best circumstances. Based on the principles of Box–Behnken experimental design (BBD), 17 experiments were designed for the three factors at three levels: L-Met concentration (g/L), incubation time (h), and yeast extract concentration (g/L). The experimental results showed that the yield of 3-Met under different fermentation conditions ranged from 1.46 to 3.54 g/L, with the highest yield at the central values, averaging 3.43 g/L. Through multivariate regression analysis and fitting of a multivariate quadratic equation, the regression equation obtained was as follows:Y = −28.76 + 0.6451 × A + 0.695 × B + 25.28 × C − 0.005102 × A^2^ − 0.08619 × B^2^ − 14.869 × C^2^ + 0.00010 × AB − 0.0010 × AC − 0.0562 × BC

Through analysis of variance (Table 8), the simulated equation had an *F* value of 239.27, corresponding to a *p* value < 0.0001, indicating that the model was significant. The corresponding coefficient of determination *R^2^* for the regression equation was 0.9968, indicating that the established model could explain the response value well. In addition, the lack of fit was not significant, indicating a good correlation between the actual and predicted values. The model could be used to predict the yield of 3-Met. Through significance analysis, it was found that in the regression equation, the first-order terms A (yeast extract concentration), B (L-Met concentration), and C (incubation time) were all significant, indicating a significant linear effect of the three factors on the yield of 3-Met, consistent with the PB test results. However, the order of significance was different, with L-Met concentration > incubation time > yeast extract concentration in the RSMD. This may be due to the different number of factors and calculation methods in the two experimental designs. There was no significant interaction between the three factors, indicating that the interaction effect between the three factors on the yield of 3-Met by *S. cerevisiae* Y03401 was not significant. The quadratic terms A^2^, B^2^, and C^2^ were all significant, indicating that there was a surface effect of yeast extract concentration, L-Met concentration, and incubation time on the yield of 3-Met by *S. cerevisiae* Y03401. Based on the regression equation, a corresponding response surface three-dimensional graph was drawn (Figure 4), which showed that with the increase of the three factors, the yield of 3-Met first increased and then decreased. According to the fitted regression equation, when the yeast extract concentration was 0.8 g/L, L-Met concentration was 3.8 g/L, and incubation time was 63 h, the yield of 3-Met by *S. cerevisiae* Y03401 was the highest, predicted to be 3.57 g/L. This was verified by experiments conducted under this condition, and the yield of 3-Met was 3.66 g/L, which was close to the predicted value of 3.57 g/L, indicating that the model could predict the optimal conditions for the yield of 3-Met well. The amount of 3-Met produced by *S. cerevisiae* Y03401 was higher than that reported for *H. burtonii* YHM-G, but lower than that reported for *S. fibuligera* Y1402 which was currently the highest reported natural strain for the production of 3-Met [20,21]. In addition, it should be noted that compared with *H. burtonii* YHM-G, *S. cerevisiae* Y03401 had relatively weak tolerance to 3-Met, but its yield was higher than that of the former, indicating that there was a certain correlation between the ability of the strain to produce 3-Met and its tolerance to 3-Met, but there was no proportional relationship [21].

## 4. Conclusions

This article first screened the yeast strains preserved by our research team for the production of 3-Met, and selected a yeast strain Y03401 with high yield of 3-Met, which was identified as a *S. cerevisiae*. Subsequently, SFD, PB tests, SAPD, and RSMD were used to optimize the fermentation conditions for the production of 3-Met. The results showed that the highest yield of 3-Met was produced, reaching 3.66 g/L, when the glucose concentration was 60 g/L, yeast extract concentration was 0.8 g/L, and L-Met concentration was 3.8 g/L. The initial pH was 4, the incubation period was 63 h, the inoculum size was 1.6%, the shaking speed was 150 rpm, the loading volume was 50 mL/250 mL, and the temperature was 26 °C. This was one of the few natural strains that produce 3-Met with a yield of more than 3 g/L at present, providing theoretical reference for its application in alcoholic beverages such as liquor. In addition, it is necessary to conduct further in-depth research on the mechanism of the high yield of 3-Met in the absence of L-Met.

## Figures and Tables

**Figure 1 foods-13-01296-f001:**
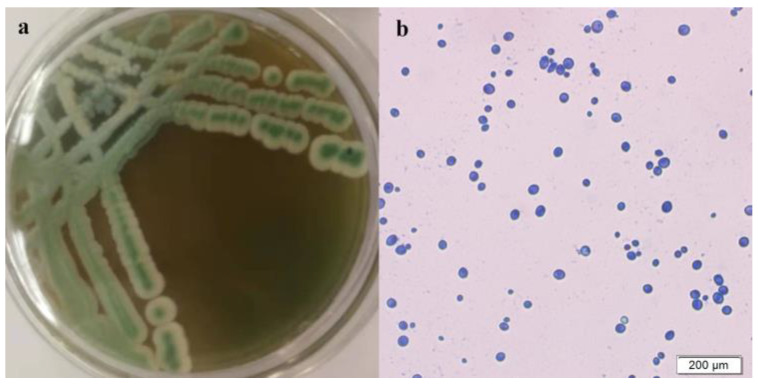
Colony morphology (**a**) on WL medium and morphological characteristics of yeast colonies under microscope (**b**).

**Figure 2 foods-13-01296-f002:**
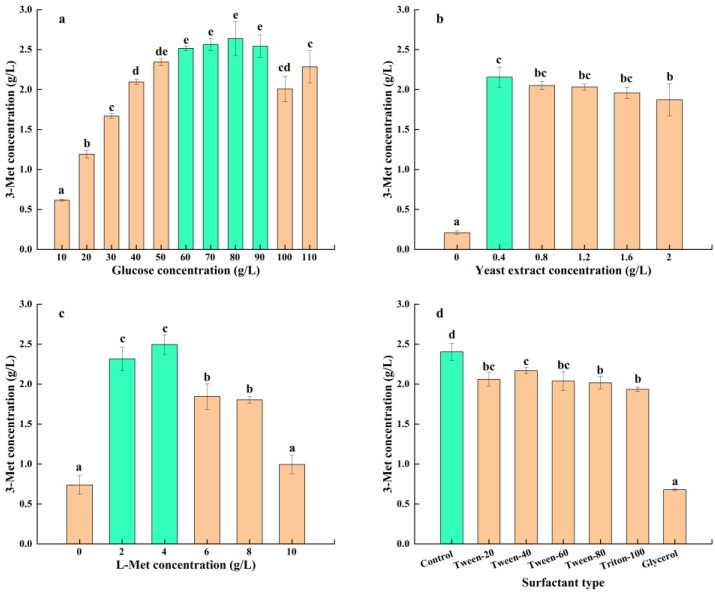
(**a**) Glucose concentration (10, 20, 30, 40, 50, 60, 70, 80, 90, 100 and 110 g/L), (**b**) yeast extract concentration (0, 0.4, 0.8, 1.2, 1.6 and 2 g/L), (**c**) L-Met concentration (0, 2, 4, 6, 8 and 10 g/L) and (**d**) surfactant types (Control, Tween-20, Tween-40, Tween-60, Tween-80, Triton-100 and Glycerol). The same letters in the column indicate that the data do not differ significantly at 5% probability using Tukey’s test. The green bar means the best conditions on 3-Met concentration.

**Figure 3 foods-13-01296-f003:**
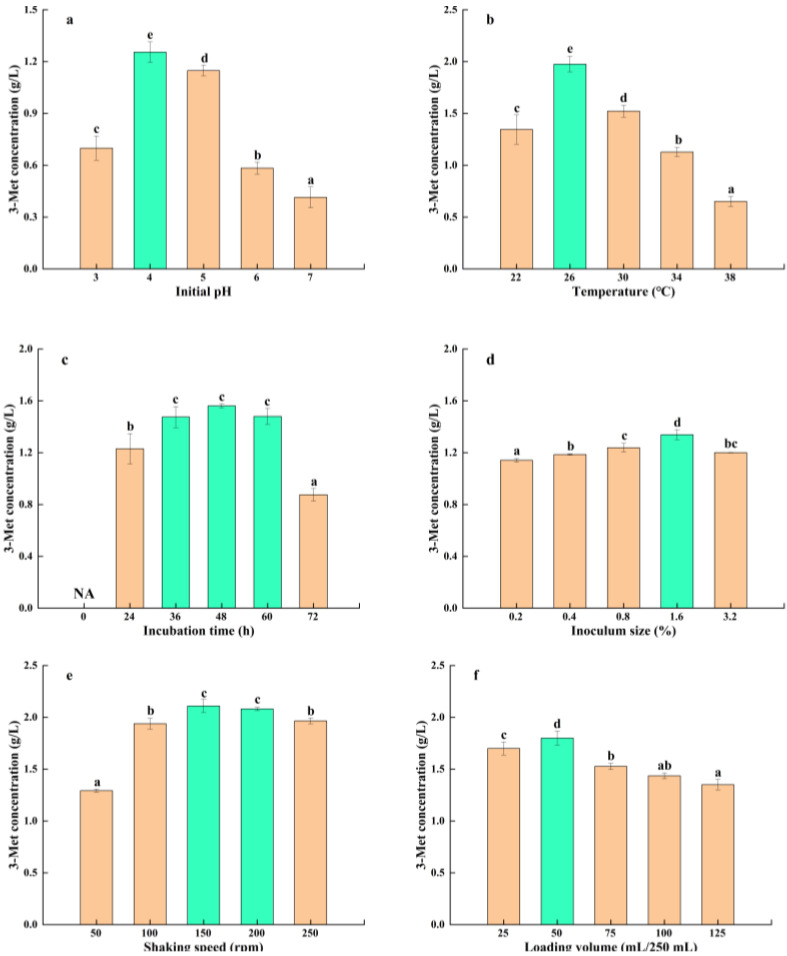
(**a**) initial pH (3, 4, 5, 6 and 7), (**b**) temperature (22, 26, 30, 34 and 38 °C), (**c**) incubation time (0, 24, 36, 48, 60 and 72 h), (**d**) inoculum size (0.2, 0.4, 0.8, 1.2 and 3.2%), (**e**) shaking speed (50, 100, 150, 200 and 250 rpm), (**f**) loading volume (25, 50, 75, 100 and 125 mL/250 mL). According to Tukey’s test, the identical letters in the column indicate there is no significant difference in the data with a 5% probability. The optimal 3-Met concentration levels are shown by the green bar. The letter NA stands for “no 3-Met detection”.

**Figure 4 foods-13-01296-f004:**
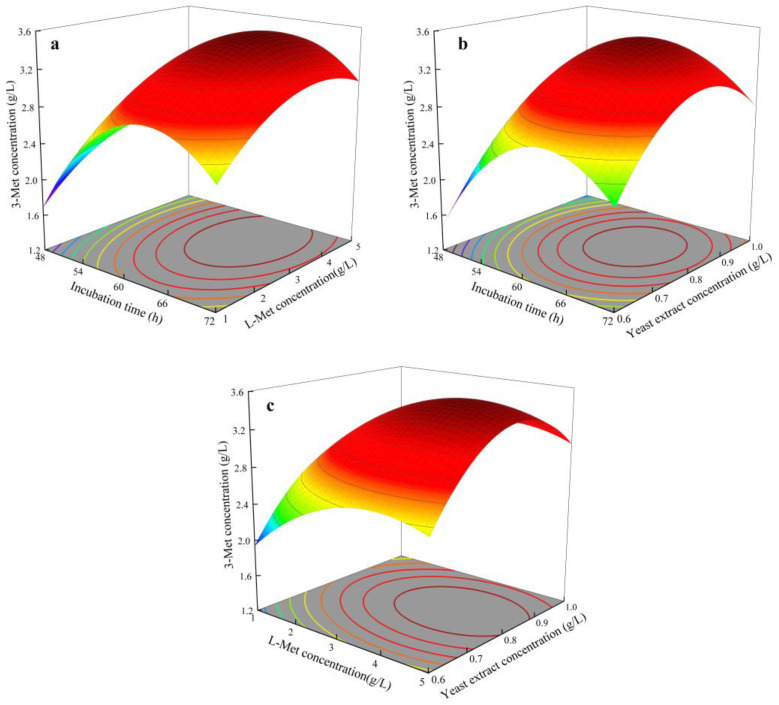
Response surface and contour map of the interaction of various factors affecting the yield of 3-Methylthiopropanol. (**a**) interaction between incubation time and L-Met concentration; (**b**) interaction between incubation time and yeast extract concentration; (**c**) interaction between L-Met concentration and yeast extract concentration. The colour change from blue to red corresponds to increase in responses.

**Table 1 foods-13-01296-t001:** Factors and levels of SFD and their optimization conditions.

Factor	Level/Type	Optimization
Glucose concentration (g/L)	10, 20, 30, 40, 50, 60, 70, 80, 90, 100 and 110	60–90
Yeast extract concentration (g/L)	0, 0.4, 0.8, 1.2, 1.6 and 2	0.4
L-Met concentration (g/L)	0, 2, 4, 6, 8 and 10	2–4
Surfactant types	Control, Tween-20, Tween-40, Tween-60, Tween-80, Triton-100 and Glycerol	Control
Initial pH	3, 4, 5, 6 and 7	4
Temperature (°C)	22, 26, 30, 34 and 38	26
Incubation time (h)	0, 24, 36, 48, 60 and 72	36–60
Inoculum size (%)	0.2, 0.4, 0.8, 1.6 and 3.2	1.6
Shaking speed (rpm)	50, 100, 150, 200 and 250	150–200
Loading volume (mL/250 mL)	25, 50, 75, 100 and 125	50

**Table 2 foods-13-01296-t002:** Plackett–Burman test design and results.

Test Number	X_1_ (g/L)	X_2_ (g/L)	X_3_ (g/L)	X_4_ (°C)	X_5_	X_6_ (h)	X_7_ (rpm)	X_8_ (mL/250 mL)	X_9_ (%)	3-Met Concentration (g/L)
1	50	0.1	4	28	5	24	200	75	0.8	0.94
2	60	0.4	2	26	4	36	150	50	1.6	1.73
3	50	0.7	4	28	3	48	200	25	2.4	3.1
4	60	0.4	2	26	4	36	150	50	1.6	1.55
5	70	0.1	4	24	3	24	200	75	2.4	1.19
6	70	0.7	4	24	5	48	100	75	0.8	2.37
7	70	0.7	0	28	5	24	200	25	0.8	1.52
8	50	0.1	0	28	5	48	100	75	2.4	1.02
9	70	0.1	0	24	5	48	200	25	2.4	1.12
10	50	0.7	4	24	5	24	100	25	2.4	2.21
11	50	0.7	0	24	3	48	200	75	0.8	1.51
12	60	0.4	2	26	4	36	150	50	1.6	1.74
13	50	0.1	0	24	3	24	100	25	0.8	0.77
14	70	0.1	4	28	3	48	100	25	0.8	1.30
15	70	0.7	0	28	3	24	100	75	2.4	1.48

**Table 3 foods-13-01296-t003:** Experimental designs and the results of SAPD.

Groups	1	2	3	4	5
Incubation time (h)	24	36	48	60	72
L-Met concentration (g/L)	0	1	2	3	4
Yeast extract concentration (g/L)	0.2	0.4	0.6	0.8	1
3-Met concentration (g/L)	1.08	1.78	2.37	3.29	1.03

**Table 4 foods-13-01296-t004:** The RSMD and the responses of the dependent variables.

Test Number	Incubation Time (h)		L-Met Concentration (g/L)		Yeast Extract Concentration (g/L)		3-Met Concentration (g/L)
	A	Code A	B	Code B	C	Code C	Y
1	60	0	3	0	0.8	0	3.37
2	72	1	3	0	0.6	−1	2.24
3	60	0	3	0	0.8	0	3.35
4	72	1	3	0	1	1	2.74
5	60	0	3	0	0.8	0	3.45
6	48	−1	3	0	0.6	−1	1.46
7	48	−1	1	−1	0.8	0	1.69
8	60	0	1	−1	0.6	−1	1.94
9	48	−1	3	0	1	1	1.97
10	72	1	5	1	0.8	0	3.02
11	60	0	3	0	0.8	0	3.45
12	72	1	1	−1	0.8	0	2.46
13	60	0	1	−1	1	1	2.49
14	60	0	5	1	0.6	−1	2.54
15	48	−1	5	1	0.8	0	2.24
16	60	0	3	0	0.8	0	3.54
17	60	0	5	1	1	1	3.00

**Table 5 foods-13-01296-t005:** Yield of 3-Met by various strains.

Strain	3-Met Concentration (g/L)	Strain	3-Met Concentration (g/L)	Strain	3-Met Concentration (g/L)
1021	0.78 ± 0.06	F12502	0.31 ± 0.09	F1918	0.37 ± 0.06
1213	0.31 ± 0.11	F13004	0.58 ± 0.04	F1915	0.55 ± 0.07
1273	0.46 ± 0.03	F13011	0.00 ± 0.00	F2405	0.00 ± 0.00
1344	0.21 ± 0.07	F13017	0.00 ± 0.0	Y03401	1.30 ± 0.06
1701	0.51 ± 0.03	F1504	0.77 ± 0.08	Y1401	0.00 ± 0.00
32704	0.57 ± 0.10	F1701	0.21 ± 0.05	Y1405	0.66 ± 0.13
32708	0.48 ± 0.08	F1702	0.00 ± 0.00	Y1511	0.27 ± 0.06
32714	0.43 ± 0.01	F1904	0.36 ± 0.06	Y1517	0.00 ± 0.00
33197	0.59 ± 0.07	F1905	0.00 ± 0.00	Y2#09	0.38 ± 0.02
33260	0.36 ± 0.02	F1912	0.00 ± 0.00	Y41114	0.45 ± 0.06
33298	0.76 ± 0.11	F1913	0.33 ± 0.08	Y8#01	1.10 ± 0.10
33319	0.92 ± 0.06	F1914	0.77 ± 0.11	Y8#03	0.96 ± 0.08
F10404	1.21 ± 0.08	F1916	0.52 ± 0.04	YX3307	0.39 ± 0.05

**Table 6 foods-13-01296-t006:** Physiological, biochemical characteristics and performance test results of *S. cerevisiae* Y03401.

Tests	Substrate Type	Results	Tests	Substrate Type	Results
Nitrogen source assimilation tests	Urea	+	Sugar fermentation tests	D-Xylose	Gas production, no acid
	Ammonium sulfate	+		Fructose	Acid and gas production
	Potassium nitrate	+		Glucose	Acid and gas production
	L-Met	+		Maltose	Acid and gas production
	L-Tyrosine	+		Lactose	Gas production, no acid
Carbon source assimilation tests	Ethanol	+		Sucrose	Acid and gas production
	Glycerol	+	Other tests	Hydrogen sulfide test	−
	Inulin	+		Indole test	−
	D-Ribose	−		Methyl red test	+
Tolerance range	Temperature (°C)	20–40		Voges–Proskauer test	−
	NaCl (%, *w*/*v*)	0–6		Citrate test	+
	Ethanol (%, *v*/*v*)	0–6		Urea test	−
	Glucose (%, *w*/*w*)	0–45.9		Starch hydrolysis test	+
	pH	2–10		Gelatin liquidized test	−
	3-Met (g/L)	1–12			

Note: “+”, positive response; “−”, negative response.

**Table 7 foods-13-01296-t007:** Levels of the variables and statistical analysis in PB for yield of 3-Met.

Code	Variable	Low Level (−1)	High Level (+1)	Effect (EXi)	*F* Values	*p* Values	Rank	Significance
X_1_	Glucose concentration (g/L)	50	70	−0.0491	0.45	0.532	6	-
X_2_	Yeast extract concentration (g/L)	0.1	0.7	0.4859	44.1	0.001	1	***
X_3_	L-Met concentration (g/L)	0	4	0.3081	17.72	0.008	2	**
X_4_	Temperature (°C)	24	28	0.0159	0.05	0.837	8	-
X_5_	Initial pH	3	5	−0.0146	0.04	0.849	9	-
X_6_	Incubation time (h)	24	48	0.1922	6.9	0.047	3	*
X_7_	Shaking speed (rpm)	100	200	0.0203	0.08	0.792	7	-
X_8_	Loading volume (mL/250 mL)	25	75	−0.1256	2.95	0.147	5	-
X_9_	Inoculum size (%)	0.8	3.2	0.1436	3.85	0.107	4	-
	model				8.46	0.015		*
	*R*^2^ = 0.9384		*R*^2^_Adj_ = 0.8274					

Note: “-” Not significant (*p* > 0.05); “*” Significant at 5% level (*p* < 0.05); “**” Significant at 1% level (*p* < 0.01); “***” Significant at 0.1% level (*p* < 0.001).

**Table 8 foods-13-01296-t008:** Regression coefficients and their significance for 3-Met yield from the results of the BBD.

Source	Sum of Squares	DF	Mean Square	*F* Values	*p* Values
Model	7.03873	9	0.78208	239.27	0.000
A-Yeast extract concentration	0.02282	1	0.02282	6.98	0.033
B-L-Met concentration	0.34501	1	0.34501	105.55	0.000
C-Incubation time	0.09187	1	0.09187	28.11	0.001
AB	0.00203	1	0.00203	0.62	0.457
AC	0.00003	1	0.00003	0.01	0.933
BC	0.00002	1	0.00002	0.01	0.933
A^2^	1.48938	1	1.48938	455.67	0.000
B^2^	0.50043	1	0.50043	153.10	0.000
C^2^	2.27308	1	2.27308	695.44	0.000
Residual	0.02288	7	0.00327		
Lack of Fit	0.00000	3	0.00000	0.00	1.000
Pure Error	0.02288	4	0.00572		
Cor Total	7.06161	16			
	*R*^2^ = 0.9968	*R*^2^_Adj_ = 0.9926			

## Data Availability

The original contributions presented in the study are included in the article, further inquiries can be directed to the corresponding authors.

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
