# Peer review of "Screening of a Saccharomyces cerevisiae Strain with High 3-Methylthio-1-Propanol Yield and Optimization of Its Fermentation Conditions"

_foods, 2024, doi:10.3390/foods13091296_

Round 1
Reviewer 1 Report
Comments and Suggestions for Authors
The paper entitled “Screening of a Saccharomyces cerevisiae Strain with High 3-Me-thylthio-1-propanol Yield and Optimization of Its Fermentation Conditions” is focused on screening relevant functional microorganisms from brewing foods, in order to use these microorganisms to increase the content of 3-Met in alcoholic beverages and improve their quality.
The aims of the study are clearly expressed.
The experimental program is described in such manner that it can be easily applied.
The obtained results are concise, presented and discussed.
Conclusions are drawn according to the obtained data.
I congratulate the authors for their efforts in writing the paper. After minor corrections in text editing it can be accepted for publication.
Comments on the Quality of English Language
Minor editing of English language required
Author Response
202404011
Dear Reviewer,
Re: Manuscript ID. Foods-2946777 “Screening of a Saccharomyces cerevisiae Strain with High 3-Methylthio-1-propanol Yield and Optimization of Its Fer-mentation Conditions”
Thank you very much for your interest in our manuscript, and for providing us with helpful suggestions for improving its quality. In the following, we have made modifications and replies point-by-point according to your suggestions. Revisions in the manuscript are highlighted in blue.
We look forward to hearing from you at your early convenience.
Yours sincerely,
Guangsen Fan
Address: School of Food and Health, Beijing Technology and Business University, Beijing 100048, China
Tel: +86 13811497684
E-mail: fanguangsen@btbu.edu.cn
Responses to the comments
- The paper entitled “Screening of a Saccharomyces cerevisiae Strain with High 3-Me-thylthio-1-propanol Yield and Optimization of Its Fermentation Conditions” is focused on screening relevant functional microorganisms from brewing foods, in order to use these microorganisms to increase the content of 3-Met in alcoholic beverages and improve their quality.
Response: Dear Reviewer, thank you very much for your recognition of our manuscript. Our team has recently been working mainly on screening functional microorganisms that produce Baijiu-specific flavor substances or precursor substances from Baijiu brewing environment, so as to provide excellent strains for the intelligent manufacturing of Baijiu in the future. However, this kind of research is indeed very hard and difficult to form innovative articles. Therefore, we are particularly grateful for your support and recognition of our work.
- The aims of the study are clearly expressed.
Response: Thank you very much for your affirmation of our manuscript.
- The experimental program is described in such manner that it can be easily applied.
Response: Thank you very much for your agreement with our description of the experimental methods.
- The obtained results are concise, presented and discussed.
Response: Dear Reviewer, thank you very much for your positive evaluation of our results.
- Conclusions are drawn according to the obtained data.
Response: Dear Reviewer, thank you very much for your recognition of our conclusion summary.
- I congratulate the authors for their efforts in writing the paper. After minor corrections in text editing it can be accepted for publication.
Response: Dear Reviewer, thank you very much for your advice. We have revised the language of the manuscript. Thank you for your positive evaluation and approval of our manuscript.

Reviewer 2 Report
Comments and Suggestions for Authors
An interesting manuscript on the optimization of the 3-Met production by a S. cerevisiae strain. The significance of 3-Met production lies in the fact that it is a very important flavor compound in alcoholic beverages. However, the authors failed to discuss the relevance of the optimized parameters and conditions in relation to the actual production procedure, at least of the alcoholic beverages they mention, namely Baijiu and Huangjiu. This will highlight the practical significance of the study.
The following also need attention:
l. 54-58. ‘Unfortunately…process’ please rephrase to improve clarity.
l. 80. What is meant by the word ‘saved’?
paragraph 2.2. Please explain the rather weird sterilization conditions. The typical conditions are 121 oC, 15 psi, 15 min. Why do the authors oversterilize their media; doesn’t this affect negatively their performance?
l. 122. Please report preservation conditions.
l. 124, 354-363, 387, and possibly elsewhere what is meant by ‘incubation size’?
l. 125. Please report the g value
l. 134. The classical identification was performed; at least a reference on how the tests were performed should be given
table 1. ‘Triton’
table 5. An ANOVA and indication of the statistically important differences would help differentiate the producing capacities of the strains. Please mention it also in the materials and methods.
l. 218. Please report the ‘certain primers’
l. 220-221. ‘It… traits’ Which strain do the authors refer to?
l. 230-231. ‘which…beverages’. What is the alcohol level of ‘Baijiu and other alcoholic beverages’?
l. 250. What is meant by word ‘germs’?
l. 252. Please remove the declension from the number
l. 297. Please specify which tween and which triton
figures 2, 3. What is represented by the letters above each bar. Please describe it also in the materials and methods.
l. 315. They have not been removed, therefore ‘are shown’
l. 340-342. ‘Therefore…3-Met’ please rephrase to improve clarity.
l. 356, 563, 566, 595 please write the scientific names in italics
Author Response
20240409
Dear Reviewer,
Re: Manuscript ID. Foods-2946777 “Screening of a Saccharomyces cerevisiae Strain with High 3-Methylthio-1-propanol Yield and Optimization of Its Fer-mentation Conditions”
Thank you very much for your interest in our manuscript, and for providing us with helpful suggestions for improving its quality. In the following, we have made modifications and replies point-by-point according to your suggestions. Revisions in the manuscript are highlighted in blue.
We look forward to hearing from you at your early convenience.
Yours sincerely,
Guangsen Fan
Address: School of Food and Health, Beijing Technology and Business University, Beijing 100048, China
Tel: +86 13811497684
E-mail: fanguangsen@btbu.edu.cn
Responses to the comments
- An interesting manuscript on the optimization of the 3-Met production by a S. cerevisiae strain. The significance of 3-Met production lies in the fact that it is a very important flavor compound in alcoholic beverages. However, the authors failed to discuss the relevance of the optimized parameters and conditions in relation to the actual production procedure, at least of the alcoholic beverages they mention, namely Baijiu and Huangjiu. This will highlight the practical significance of the study.
Response: Dear Reviewer, how are you? Thank you very much for your recognition of our manuscript and for your valuable suggestions, which are of great help to improve the quality of our manuscript. Your advice on combining production parameters of Baijiu and Huangjiu is very beneficial, which coincides with our current work on solid-state simulated fermentation of the strain to enhance Baijiu flavor. We are exploring the function of this strain in enhancing Daqu and improving Baijiu flavor, mainly through analyzing the changes in microbial flora, flavor, and physicochemical indicators. The main purpose of this article is to explore the potential of this strain to produce 3-methylthiopropanol under optimal fermentation conditions, and only some optimized conditions, such as temperature and pH, may be appropriately discussed and analyzed in relation to actual parameters related to Baijiu brewing production.
Page 10, line 324-330:
This might be attributed to the fact that, under these initial pH conditions, the capacity of S. cerevisiae Y03401 to absorb nutrients or the functionality of enzymes and proteins involved in the Ehrlich pathway was relatively impaired, as evidenced by its optimal growth pH of 4.0. Additionally, this was similar to the optimal pH for 3-Met produc-tion by H. burtonii YHM-G, which also originated from Baijiu, and this might be related to the slightly acidic environment in which both were used for Baijiu production [21, 29].
Page 10, line 342-344:
When the temperature was 26 ℃, the yield reached its maximum, which was consistent with its optimal growth temperature and slightly lower than the reported optimal temperature for other yeast strains producing 3-Met [14, 21].
The following also need attention:
- 54-58. ‘Unfortunately…process’ please rephrase to improve clarity.
Response: Thank you very much for your advice. With your reminder, we have reformulated the sentence as follows:
Page 2, line 54-56:
Unfortunately, at present, these alcohol products generally have a low content of 3-Met during the brewing process, which affects the quality of the related products to a certain extent
- 80. What is meant by the word ‘saved’?
Response: We are very sorry, but there was a mistake in our previous statement. We have made the necessary corrections.
Page 2, line 80-81:
The experiment's yeast strains were selected from several aroma-type Baijiu brewing environments and then preserved by our research team.
- paragraph 2.2. Please explain the rather weird sterilization conditions. The typical conditions are 121 oC, 15 psi, 15 min. Why do the authors oversterilize their media; doesn’t this affect negatively their performance?
Response: Thank you very much for your advice. We have followed the existing research reports for the preparation and sterilization of the culture medium. We have also noticed the issue of nutrient retention. For unstable components in the culture medium, we will use a membrane to remove possible contaminants, sterilize sugar substances separately before mixing them with other sterilized ingredients, and reduce the corresponding sterilization temperature and shorten the time. After consulting literature and books, we found that the commonly used sterilization parameters for microbial culture media in our country are either 121°C for 20 minutes or 115°C for 30 minutes. We appreciate the slight difference in the sterilization conditions you mentioned. Additionally, we are very grateful for your advice, which also allowed us to discover a mistake in the description of the sterilization conditions, and we have corrected it. In future research, we will pay more attention to the sterilization conditions of the culture medium and make careful choices.
We have listed examples of literature published this year that adopted the sterilization parameters of 121°C for 20 minutes.
(1) Ding, Z., Zhang, L., Xu, Z. et al. Isolation of a marine-derived yeast with potential applications in industrial nitrite utilizing. 3 Biotech 14, 29 (2024). https://doi.org/10.1007/s13205-023-03866-8
(2) Zhu, L., Zhang, X., Wang, Y. et al. Recovery and characterization of β-glucosidase-producing non-Saccharomyces yeasts from the fermentation broth of Vitis labruscana Baily × Vitis vinifera L. for investigation of their fermentation characteristics. Arch Microbiol 206, 174 (2024). https://doi.org/10.1007/s00203-024-03878-9
(3) Liu, X.; Yang, W.; Gu, H.; Bughio, A.A.; Liu, J. Optimization of fermentation conditions for 2,3,5-trimethylpyrazine produced by Bacillus amyloliquefaciens from Daqu. Fermentation 10, 112 (2024). https://doi.org/10.3390/fermentation10020112
(4) Ryu, K.M., Kim, H., Woo, J. et al. Enhancement of the bioactive compounds and biological activities of maca (Lepidium meyenii) via solid-state fermentation with Rhizopus oligosporus. Food Sci Biotechnol (2024). https://doi.org/10.1007/s10068-023-01508-6
(5) de Oliveira Pereira, I., dos Santos, Â. A., Guimarães, N. C., Lima, C. S., Zanella, E., Matsushika, A., Rabelo, S. C., Stambuk, B. U., & Ienczak, J. L. First- and second-generation integrated process for bioethanol production: fermentation of molasses diluted with hemicellulose hydrolysate by recombinant Saccharomyces cerevisiae. Biotechnol Bioeng 121, 1314–1324, (2024). https://doi.org/10.1002/bit.28648
- 122. Please report preservation conditions.
Response: Thank you very much for your advice. We have supplemented the relevant information accordingly.
Page 3, line 122-123:
The yeast strains preserved in glycerin tube at -20 ℃ were inoculated into YPD me-dium and activated for 48 hours at 28 ℃ and 180 rpm.
- 124, 354-363, 387, and possibly elsewhere what is meant by ‘incubation size’?
Response: We are very sorry for the confusion. What we meant was the inoculum size. We have corrected the relevant errors in the text.
- 125. Please report the g value
Response: Thank you very much. We have made the necessary modifications based on your suggestions.
- 134. The classical identification was performed; at least a reference on how the tests were performed should be given
Response: Thank you very much for your feedback. We referred to the classic identification method, and based on your advice, we have supplemented the relevant references.
- table 1. ‘Triton’
Response: Thank you very much for your careful attention. We have made the necessary modifications.
- table 5. An ANOVA and indication of the statistically important differences would help differentiate the producing capacities of the strains. Please mention it also in the materials and methods.
Response: Thank you very much for your suggestion. We have supplemented the description of the relevant statistical methods in the section of Materials and Methods.
Page 5, line 178-179:
A one-way ANOVA (p < 0.05) with Tukey's test was used to analyze statistical differences in the assessed methodologies.
- 218. Please report the ‘certain primers’
Response: Thank you very much for your feedback. We have supplemented the relevant information accordingly.
Page 6, line 219-221:
After extracting the whole DNA from yeast Y03401, the 26s rDNA gene fragment was successfully amplified using specific primers, NL1 (5′-GCATATCAATAAGCGGAGGAAAAG-3′) and NL4 (5′-GGTCCGTGTTTCAAGACGG-3′).
- 220-221. ‘It… traits’ Which strain do the authors refer to?
Response: Thank you very much. It specifically refers to two strains of Saccharomyces cerevisiae, S. cerevisiae Y1914 and Y3401, which are both derived from the Baijiu brewing system.
- 230-231. ‘which…beverages’. What is the alcohol level of ‘Baijiu and other alcoholic beverages’?
Response: Thank you very much for your advice. We have supplemented the relevant information.
Page 7, line 232-235:
- cerevisiae Y03401 had moderate ethanol tolerance and grew well when the ethanol volume fraction is 6% or lower, which made it possible to adjust to the 6% or less eth-anol content that is typically produced in the brewing of Baijiu and some other alco-holic beverages process [32, 33].
- 250. What is meant by word ‘germs’?
Response: Thank you very much for pointing out our shortcomings. We meant microorganisms, and we have made the corresponding replacement.
Page 8, line 253-256:
However, since various microbes have varying glucose tolerances, higher glucose concentration wasn't always preferable for the development and metabolism of microbes; rather, it was helpful in specific circumstances [35].
- 252. Please remove the declension from the number
Response: Thank you very much. We have made deletions according to your advice.
Page 6, line 219-221:
After extracting the whole DNA from yeast Y03401, the 26s rDNA gene fragment was successfully amplified using specific primers, NL1 (5′-GCATATCAATAAGCGGAGGAAAAG-3′) and NL4 (5′-GGTCCGTGTTTCAAGACGG-3′).
Page 8, line 256-260:
The influence of glucose concentration on the production of 3-Met by S. cerevisiae Y03401 is clearly depicted in Figure 3a. The figure showed that when glucose concen-tration increased, the demand for energy and carbon sources necessary for the growth, metabolism, and reproduction of S. cerevisiae Y03401 escalated progressively.
Page 10, line 324-327:
This might be attributed to the fact that, under these initial pH conditions, the capacity of S. cerevisiae Y03401 to absorb nutrients or the functionality of enzymes and proteins involved in the Ehrlich pathway was relatively impaired, as evidenced by its optimal growth pH of 4.0.
Page 11, line 376-380:
Therefore, it was crucial to investigate how these two variables influenced the capabil-ity of S. cerevisiae Y03401 to produce 3-Met. At lower shaking speeds (50 rpm), the flask's oxygen level was insufficient, adversely affecting the growth and fermentation of S. cerevisiae Y03401, and subsequently reducing the production of 3-Met, as depicted in Figure 4e.
- 297. Please specify which tween and which triton
Response: Thank you very much for your feedback. This part only provides a general description of Tween and Triton without specific explanations, as the optimal surfactant type may vary for different microorganisms, and even the same microorganism may require different surfactants when producing different metabolites.
- figures 2, 3. What is represented by the letters above each bar. Please describe it also in the materials and methods.
Response: Thank you very much for your advice. We have made supplements in the section of Materials and Methods.
Page 5, line 178-179:
A one-way ANOVA (p < 0.05) with Tukey's test was used to analyze statistical differences in the assessed methodologies.
Page 9, line 314-315:
The same letters in the column indicate that the data do not differ significantly at 5% probability using Tukey’s test. The green bar means the best conditions on 3-Met concentration.
Page 11, line 393-396:
According to Tukey's test, the identical letters in the column indicate there is no significant difference in the data with a 5% probability. The optimal 3-Met concentration levels are shown by the green bar. The letter NA stands for "no 3-Met detection".
- 315. They have not been removed, therefore ‘are shown’
Response: Thank you very much for pointing out this issue in our manuscript. We have made the necessary modifications based on your advice.
Page 10, line 321-322:
The results of the initial pH's effect on the yield of 3-Met by S. cerevisiae Y03401 are shown in Figure 4a.
- 340-342. ‘Therefore…3-Met’ please rephrase to improve clarity.
Response: Thank you very much for your comments. We have rewritten this part of the content as follows: Therefore, it was crucial to investigate how these two variables influenced the capability of S. cerevisiae Y03401 to produce 3-Met. At lower shaking speeds (50 rpm), the flask's oxygen level was insufficient, adversely affecting the growth and fermentation of S. cerevisiae Y03401, and subsequently reducing the production of 3-Met, as depicted in Figure 4e. (Page 11, line 376-380)
- 356, 563, 566, 595 please write the scientific names in italics
Response: Thank you very much for your comments. We apologize for the mistake in the writing format of the manuscript and have checked the writing of microbes names in the whole text.
Page 8, line 251:
Effect of medium components on 3-Met production by S. cerevisiae Y03401
Page 10, line 316:
Effect of fermentation conditions on 3-Met production by S. cerevisiae Y03401
Page 10, line 364-365:
The quantity of 3-Met generated by S. cerevisiae Y03401 somewhat increased as the inoculum size increased, as Figure 4d illustrates.

Round 2
Reviewer 2 Report
Comments and Suggestions for Authors
I have checked the manuscript, the authors have addressed all the comments and the manuscript can be published.